# Derek Jarman's *Tempest*, William Shakespeare's *Salò*

**Tomas Elliott**

Department of English, Northeastern University London, London E1W 1LP, UK; tomas.elliott@nulondon.ac.uk

**Abstract:** This article re-evaluates Derek Jarman's adaptation of William Shakespeare's *The Tempest* (1979) based on archival research into the cinematic and historical intertexts that influenced the film. Specifically, it focuses on the impact of Pier Paolo Pasolini on Jarman's aesthetics, particularly the Italian filmmaker's last work: *Salò, or the 120 Days of Sodom* (1975). The article explores how Jarman used Pasolini's work as a filter through which to frame his adaptation of Shakespeare's play. In so doing, he produced a decidedly Pasolinian twist on *The Tempest*, which he explicitly referred to in his notes as "Shakespeare's *Salò*." Bridging the gap between the Renaissance and Jarman's contemporary moment, Jarman's film offers a meditation on ideas of captivity and captivation in *The Tempest*, which extends from the play and film's literal representations of imprisonment to their exploration of the affective power of performance and spectacle.

**Keywords:** Derek Jarman; Pier Paolo Pasolini; William Shakespeare; *The Tempest*; adaptation

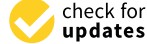



## 1. Introduction

Derek Jarman spent many years preparing his adaptation of William Shakespeare's *The Tempest* before it eventually appeared in 1979. He engaged in meticulous research, steeped in his "readings in the Renaissance magi—Dee, Paracelsus, Fludd, and Cornelius Agrippa," which, as he put it in his memoir *Dancing Ledge*, helped him to "conjure the film" (Jarman 1991a, p. 180). His understanding of and engagement with early modern magic was profound, and he set out to explore how he might transform the magic of Shakespeare's stage into the magic of cinema. Indeed, he even viewed the art form itself in somewhat mystical terms, referring to film as the "wedding of light and matter—an alchemical conjunction" (Jarman 1991a, p. 180). At the same time, however, as well as being inspired by Renaissance theories of magic, Jarman's *Tempest* incorporated influences from twentieth-century European art cinema, including, among others, the films of Robert Wiene, Jean Cocteau, Ingmar Bergman, and, most importantly, Pier Paolo Pasolini. In his *Tempest* (1979), Jarman reframed the discourse of sympathetic magic that was incarnated in Shakespeare's Prospero and filtered it through this lineage of twentieth-century filmmakers, all of whom, he believed, had produced "films where magic works."[1] By examining the extensive archive of materials that record Jarman's preparation of his film, this article will explore this strange combination of influences, focusing particularly on the British filmmaker's debt to Pasolini. It will argue that Jarman's meditations on Pasolini's late filmmaking led to the development of a *Tempest* that was as much concerned with the affective power of cinema as with the physical power of early modern magic. Jarman's resultant film bridged the gap between the Renaissance and the modern by way of an extended reflection on the themes of captivity and captivation in *The Tempest*, themes that encompass everything from the play and film's literal representation of imprisonment to their exploration of the captivating power of performance and spectacle. Thus, in Jarman's hands, Prospero's "potent art" became both a mechanism for subjugation and an opportunity to examine the affective magic of cinema (Shakespeare 2011, 5.1.59).

The depth of Derek Jarman's engagement with Shakespeare's *Tempest* and its historical background is attested by the sheer volume of notes, film treatments, scripts, and designs that he made while developing the film. Over the course of several years, he drafted

and redrafted versions of the play, literally cutting up books of the text and gluing them together as scripts. Those scripts were then further adorned with image-cuttings, drawings, and appendices, becoming, in their own right, illuminated cinematic manuscripts in the style of the filmmaker's renowned beautiful sketchbooks. Heathcote Williams, who played Prospero in Jarman's film, once said that the filmmaker "flow[ed] with the glue," and both Jarman's method and the style of his finished version of *The Tempest* seem to bear out this assessment (Jarman 1991a, p. 188). This "flow"—Jarman's cut-and-paste methodology— allowed the filmmaker to create a collage across time, ensuring that his version of *The Tempest* was, as he put it, less a reflection on a particular period and more "a chronology of the 350 years of the play's existence, like the patina on old bronze" (Jarman 1991a, p. 188). This patina incorporated not only the background to Shakespeare's play in its own time but also its relationship to twentieth-century European art cinema, and this allowed Jarman to understand his research into early modern magic in relation to his contemporary moment. Nowhere is this better evidenced than in a "Script and Film Treatment" for *The Tempest* that Jarman produced in around 1976–1977. In this treatment, he focused extensively on the historical moment in which Shakespeare's play debuted, when it was first performed at the court of James I. He pointed to the fact that the play emerged during a pivotal period in English history, when the notions of sympathetic magic (epitomized by John Dee, Giordano Bruno, and others), along with the entire epistemological and political systems that they had inspired, were coming under attack. In this context, a surprising comparison with the present occurred to him:

> John Dee, Elizabeth's astrologer, has only just died in disgrace, and James was in the forefront of the movement to discredit the Elizabethan obsession with sympathetic magic, the world of Spenser, the large allegorical paintings of the Virgin Queen, and *Dr Faustus*, and *The Tempest*. Shakespeare in his last play, which was performed for the marriage of James' daughter Elizabeth, the elector palatine, returns for inspiration to the dream of his Elizabethan youth. Perhaps the "Winter Queen" and her young husband, like Miranda and Ferdinand, would be the inheritors of the English Renaissance. *The Tempest* was therefore a highly political play with an audience who believed in the reality of its magic. It was Shakespeare's *Salò*! ("Script": fol. 13)

The leap is undoubtedly surprising. Rowland Wymer, in his detailed assessment of Jarman's films, refers to it as "an odd remark" (2005, p. 74). As in *Jubilee* (1978), a film composed alongside *The Tempest*, which sees John Dee dragged into the world of punk rock, Jarman here transports us in one swift jump from the beginning of the seventeenth century to the end of the twentieth, making "strange bedfellows" of the English playwright William Shakespeare and the Italian filmmaker Pier Paolo Pasolini (Shakespeare 2011: 2.2.48-9). The exclamation mark that follows the comparison might signal its strangeness or suggest that it was somewhat tongue-in-cheek, but it could also indicate a moment of clarity for Jarman. In one sense, he seems to have discovered a key that might bridge the gap between the epistemological shift that—in his mind—occurred in England in the early 1600s, and the project that he himself was undertaking in the late 1970s. That key was Pasolini's last and most controversial film: *Salò o le 120 giornate di Sodoma* (*Salò, or the 120 Days of Sodom*, 1975).

In trying to puzzle out how Jarman came to view *The Tempest* as "Shakespeare's *Salò*," this article will endeavor to nuance several previous analyses of the film. These accounts have tended to debate the film's relative progressiveness (or lack thereof), assessing whether its representation of queer and minority communities is positive or negative. Thus, the film has been declared "an embarrassment" on the grounds that "the concerns of colonialism are largely absent" from it (MacCabe 1996, p. 194). It has been defended for its depiction of Caliban as "a white 'wage slave'" (Pencak 2002, p. 103). Its masque sequence has been praised for the "utopian impulse" of its representation of "camp" communities (Ellis 2009, p. 83), and that same sequence has been re-assessed for having "erroneously been described as 'camp'" (Charlesworth 2011, p. 74). These debates, though undoubtedly important, largely subordinate the film to the concerns of identity politics, assessing it in relation to

how successfully it represents a range of different, overlapping communities. I would argue, however, that these contrasting critical assessments result from the fact that the film's representation of those communities is deeply ambivalent. This makes sense if we understand the film as a dramatization of "Shakespeare's *Salò*" since Jarman seems to have valued Pasolini's film precisely for its ambivalences. Thus, by building off Pasolini's representation of violent sexuality in *Salò*, Jarman in his *Tempest* refuses simply to represent queer and minority communities as inherently and necessarily either good or bad; rather he demonstrates how all such communities can become victim to powerful imbalances and manipulations. In so doing, Jarman uses his adaptation of Shakespeare's play to explore the continuities that exist between the machinations of magical power at work on Prospero's island, the aggressive and murderous practices of the libertines in Pasolini's *Salò*, and the patterns of desire (queer or otherwise) in late twentieth-century European cinema. Echoing Pasolini in *Salò*, Jarman in *The Tempest* dramatizes relationships that seem constantly at risk of slipping into hierarchies of control; whereas in *Salò* those hierarchies are maintained primarily through violent physical abuse and through psychodynamic manipulation, in *The Tempest* they are sustained through a belief in "the reality of [the play's] magic" ("Script": fol. 13).

To unpack these lines of influence more thoroughly, I will begin with an overview of Pasolini's wide-ranging and extensive impact on Jarman before examining how this filtered specifically into his adaptation of *The Tempest*.

## 2. The Great Queer Artists Dealt with Negatives

That Pasolini's *Salò* should even have occurred to Jarman in the context of his research into John Dee and sympathetic magic suggests that the Italian filmmaker was ever-present in his mind. In many ways, Jarman saw Pasolini everywhere, making the connection he drew between *The Tempest* and *Salò* rather more typical than singular. On the very first page of *Dancing Ledge*, for example, when describing his preparations for *Caravaggio* (1986), Jarman writes: "had Caravaggio been reincarnated in this century it would have been as a filmmaker, Pasolini" (Jarman 1991a, p. 1). He seems, in other words, to have made a habit of locating Pasolini in the Renaissance past of his own filmography. Throughout the rest of *Dancing Ledge*, Jarman returns to Pasolini again and again, recalling at one point his endearing excitement at meeting Pasolini and having the opportunity to tell him "how much [he] loved his films," and also recounting a touching episode in which, at the first press screening of *Sebastiane* (1976) in Rome, the writer Alberto Moravia (who had given the eulogy at Pasolini's funeral), praised the film and said that "it was a film that Pier Paolo would have loved" (Jarman 1991a, pp. 157–58).

Many other scholars have pointed to Pasolini's influence on Jarman. Colin MacCabe (2011, p. 506) calls Pasolini "Jarman's great master," and Jim Ellis (2009, p. 33) has noted that Jarman developed a film script concerning Pasolini entitled *P.P.P. in the Garden of Earthly Delights* and even "played Pasolini late in life in Julia Cole's short film *Ostia*" (1991). This suggests that the specter of Pasolini haunted Jarman long after the former's death. Jim Ellis argues that *Sebastiane* was "clearly an homage to Pasolini," and he claims that the darkness of Pasolini's legacy hangs over *Jubilee*: "if *Sebastiane* is Jarman's version of Pasolini's *Gospel According to St Matthew* (1964)," Ellis writes, "then Jarman's next film, *Jubilee*, is perhaps his *Salò*" (Ellis 2009, p. 49). Thus, as if the lines of influence were not already tangled enough, where Jarman thought of *The Tempest* as Shakespeare's *Salò*, Ellis views *Jubilee* as Jarman's *Salò*. For his part, Jarman seems to have felt that Pasolini and *Salò* were particularly present in his filmmaking whenever he explored the darker aspects of human nature and the overlap between sex and violence. In discussing his later film, *The Last of England* (1987), for example, Jarman commented that its center was "very dark, unforgiving, like *Salò*" (Jarman 1997, p. 208). As I will argue in the concluding section of this article, this darker vision of humanity is clearly present in Jarman's adaptation of *The Tempest*, even if, as Rowland Wymer (2005, p. 74) points out, it is less visible there than in his other adaptation of a Renaissance play, *Edward II* (1991).

Whatever the exact connections that Jarman identified between Shakespeare's *Tempest* and Pasolini's *Salò*, it is undeniable that both the film and the play had a lasting effect on him. As mentioned, he developed his ideas for an adaptation of *The Tempest* over an extended period of time, and that period overlapped precisely with his first encounter with Pasolini's film, which he most likely saw at the Locarno film festival in August 1976. This was close to a year after the film's initial release in Paris, a release that had been followed, just a few days later, by Pasolini's murder in Ostia in November 1975. In January 1976, *Salò* was rejected by the British Board of Film Censors, but Jarman had the chance to see it when his *Sebastiane* was selected to screen alongside *Salò* at Locarno later that year. We do not know exactly how Jarman reacted to this initial viewing—though viewers rarely forget their first time—but his response to a censored version that he saw in London nearly fifteen years later testifies to the deep regard he felt for the film. Here is Jarman writing in his journal, *Modern Nature*, in February 1990:

> We went to see *Salò* at the Scala in MGM's murdered version. Emerged numbed into the desolate cold of King's Cross. A wave of anger came over me. The pathetic nature of British life: no Pasolini, Genet, or Barthes, no-one here really. Just *Bent* at the National with everyone congratulating themselves. I find the British Theatre tedious, the thespians of Stonewall capitalizing on their truly horrid connections with the Establishment. (Jarman 1991b, p. 238)

Jarman's reaction to this "murdered version" of Pasolini's film clearly highlights his love for the original. Moreover, his anger here highlights what he shared with Pasolini: a hatred of state power, and a distaste for the artists who collude with it. His sickened reaction to the production of Martin Sherman's *Bent* (another work exploring fascist authority and sexual control, first produced in 1979), which was only supposed to have been revived from its Royal Court roots as part of a one-night benefit for Stonewall, but which in 1990 was enjoying a run at the symbolic home of state-sponsored art, the National Theatre, highlights Jarman's disgust with an English queer art scene that he felt had betrayed its roots. When he emerged from the darkened rooms of the fading Scala theatre, having witnessed this sanitized *Salò*, he saw nothing other than a world in which the twin pillars of political power and consumer culture—those pillars that Pasolini had dedicated his life to dismantling—had become ever more thoroughly secured.

To make matters worse, it was precisely a sense of "betrayal" that, Jarman felt, Pasolini had explored in *Salò* itself. When he wrote about the film in *Dancing Ledge*, in a passage reminiscent of the anger he experienced in 1990, he focused primarily on the sadness of Pasolini's situation and on the theme of betrayal in the film:

> Romans are bad lovers and good fucks—men living together in supportive relationships few.
>
> Pier Paolo, living with his mother and hitting the streets nightly to give blow-jobs to his street boys, illustrates the situation well. Though open, his sexuality was a tortured confusion, made worse by the Communist Party's adoption of bourgeois restraint. In *Salò*, his last film, all homosexual relationships are shown as decadent, unpleasant, and power-based. At the center of the film is a significant line of betrayal. Photos of loved ones lead the inquisitors on a hunt to destroy the last vestiges of private and pure relations.
>
> At the end of the line of betrayal Pasolini exhibits a STEREOTYPE, and surely one that was not in his heart. The young soldier and the black serving-girl are found in bed together by the Fascist "masters." Standing naked and defiant the boy gives a clenched fist salute, before they are both murdered in a hail of bullets. (Jarman 1991a, p. 235; emphasis in original)

This passage seems to be the most in-depth assessment of Pasolini's film that Jarman wrote. Although the "line of betrayal" that Jarman identifies ends with the murder of a heterosexual couple, Jarman's comparison of it with Pasolini's own living situation suggests



that he believed that (homo-)sexual life could, even if lived freely, still remain a torturous experience when forced to adhere to bourgeois norms and controls. Pasolini, Jarman argues, "though open," was still living a life of "confusion," subject to regulations in a global economic system that had destroyed the "last vestiges of private and pure relations." In this system, there is neither safety nor privacy for someone living a sexually transgressive lifestyle, and it is this that prompts the "betrayal" in Pasolini's film. Hence why, according to Jarman, all the homosexual (and indeed all the sexual) relationships in *Salò* are so "unpleasant and power-based." Nevertheless, it was Pasolini's willingness to engage with the negative and more destructive aspects of sexuality that Jarman seems to have valued in his filmmaking: "The great queer artists," he wrote, "dealt with negatives, this is why Pasolini and Genet will last long after *Gay Times* is forgotten in a world of false hope and illusion fed by adverts" (Jarman 2001, p. 168). The British filmmaker sees in Pasolini (and the French artist Jean Genet, mentioned yet again in the same breath as Pasolini here) a commitment not simply to view queer sexuality as necessarily progressive, revolutionary, or freeing, and to explore how it can at times be subject to the same destructive urges and repressive codes as heterosexuality. This was precisely what Jarman had experienced with the "thespians of Stonewall"—whom he viewed as belonging to an exclusive club in collusion with the establishment—and it is presumably why Pasolini's *Salò* still resonated with him so strongly fifteen years after he first saw it.

With Pasolini infusing so much of Jarman's filmmaking, though, what can we make of the very specific connection that he drew between *The Tempest* and *Salò*? It may be that, initially, Jarman was inspired by the apparent biographical parallels between Pasolini and Shakespeare, particularly given that he seems to have adhered to the critical tendency in the 1970s to view *The Tempest* as Shakespeare's "last play" ("Script": fol. 13). Given that Jarman was working in the wake of Pasolini's murder and was therefore likely very conscious of the fact that *Salò* would be Pasolini's last film, it is perhaps only natural that he might have associated it with a play that has often been thought of as Shakespeare's farewell to the stage. On the other hand, Jarman would also have been aware that *Salò* was not intended as Pasolini's last film since it was set to form part of his "Trilogy of Death." Moreover, he would probably have been aware that the view of *The Tempest* as Shakespeare's last play was by no means universally accepted. In his notes, Jarman cites Frances Yates's celebrated book *Shakespeare's Last Plays* (Yates 1975) in a list of works that he claimed would be "vitally important to the film" ("Script": fol. 19). Though Jarman was primarily interested in Yates for her association of Prospero with John Dee, it is worth noting that Yates in that book also dismisses the idea of *The Tempest* as Shakespeare's last play, stating that both *Henry VIII* and *The Two Noble Kinsmen* were written afterwards (Yates 1975, pp. 10–11).

It is not really possible to prove whether the biographical parallels between Shakespeare and Pasolini inspired Jarman to link *The Tempest* and *Salò*. We are on more certain ground, however, when we look at the overlaps between the texts themselves. The most notable of these is the isolated nature of their respective settings. *The Tempest*, of course, takes place on a (supposedly) "desolate isle" (3.3.99), a world in which Prospero seems to exercise absolute power and where he declares himself "master of a full poor cell" (1.2.23). Pasolini's *Salò*, meanwhile, takes place in the correspondingly isolated Villa Aldini, which is entirely cut off from the outside world. As many scholars have noted, that isolation means that the space of the film comes to resemble a kind of concentration camp, which explains the reference to Mussolini's fascist state in the first part of Pasolini's title. Moreover, isolation also forms a key part of the main intertextual influence behind Pasolini's film (and the inspiration for the second part of its title), the Marquis de Sade's *120 Days of Sodom* (written in 1785 and published in 1904 as *Les 120 Journées de Sodome ou l'école du libertinage*). Olga Solovieva points out that "in *120 Days of Sodom*, Sade created a prototypical concentration camp where a group of victims are brought together with the aim of systematic sexual exploitation and ultimate annihilation" (Solovieva 2019, p. 73).

We can go still further, though, by considering *Salò*'s other major intertext, which is not mentioned in Pasolini's title but which determines the film's shape and structure: the

*Inferno* from Dante Alighieri's *Divina Commedia* (1302–1321). Brian DeGrazia (2018, p. 207) has remarked that in its movement from the exterior to the interior of the Villa Aldini, Pasolini's film mirrors Dante's journey into hell. Before entering the villa, the victims are lined up outside the villa and addressed as follows by the libertines:

> Weak, chained creatures, destined for our pleasure. I hope you don't expect to find here the ridiculous freedom granted by the outside world. You are beyond the reach of any legality. No one on earth knows you are here. As far as the world is concerned, you are already dead.

DeGrazia points out the similarity between this moment and the one that takes place before the gates of hell in Dante's poem, where the speaker reads the famous lines: "Abandon all hope, ye who enter here." The world inside Pasolini's Villa Aldini, in other words, is precisely analogous to hell, entirely cut off from the outside, not unlike Prospero's island before the shipwreck. In summary, therefore, we can see that Dante's poem, Pasolini's film, and Shakespeare's play all take place in isolated spaces in which the social order is rigorously controlled by one or more beings, wielders of a supreme kind of power (in *Salò*, the Duke, the Bishop, the Magistrate, and the President are archetypal representations of four pillars of society; in *The Tempest*, Prospero could be said to incarnate all those pillars in one person).

Turning to Jarman's adaptation, we immediately notice how his *Tempest* conjures up this sense of isolation by transplanting the play from a "lush and lusty" (2.1.55) island with "sounds and sweet airs" (3.2.149) to the dark and labyrinthine interior of Stoneleigh Abbey, a "crumbling Kafkaesque pile with its gloomy stairwells" (Harris and Jackson 1997, p. 92). Jarman's notes, moreover, seem to recall Pasolini's film quite actively, since they imagine the film taking place in "an old Palladian mansion," reminding us of *Salò*'s Italian villa ("Script": fol. 2). Moving *The Tempest* inside (exterior shots are highly limited throughout) is clearly one of the most noticeable changes that Jarman made to Shakespeare's play, and it lends the film an oppressive atmosphere that seems highly redolent of Pasolini's *Salò*. It is almost as if the prison-like structure of Prospero's "full poor cell" (1.2.23) swells to encompass the entire action in Jarman's adaptation.

This oppressive Pasolinian backdrop to *The Tempest* means that Prospero's abuses of other beings within the film feel all the more intense, and the film as a whole feels more constricted. With Pasolini in mind, we can read the relationship between Prospero and Caliban, the "abhorrèd slave" (1.2.422), as structurally similar to that between the legislating masters in *Salò* and their dehumanized victims. Caliban, of course, is regularly dehumanized, and not only by Prospero. To Trinculo he is "half a fish and half a monster" (3.2.30); he is made to lie down like a dog, and that image is literalized when he kisses Stefano's shoe. We could even say that this instrumentalization of Caliban as a dog by Stephano is, in a sense, simply an earlier version of the vivid visual metaphor in *Salò*, in which the victims are held on leashes and forced to bark for their food. In *The Tempest*, barking dogs return later on in the form of Prospero's and Ariel's spirits, used to terrorize Trinculo, Stephano, and Caliban, and this scene in Jarman's adaptation works as a horrifying nightmare that again recalls Pasolini's film, with Prospero and Ariel baying and barking through the cavernous halls of the house. Throughout Jarman's adaptation, Prospero "treats Caliban with sadistic contempt" (Harris and Jackson 1997, p. 93), and critics have pointed to the Pasolinian undertones in the film's "disturbing sexual violence... such as when the naked Ferdinand is humiliated and shackled or when, in a flashback scene, Sycorax is shown controlling Ariel with a collar and chain" (Wymer 2005, p. 74). These darker and more brutal scenes (which also feature a Pasolinian interest in the consumption of base materials—uncooked eggs and breastmilk, for instance) clearly echo the extreme aesthetics of Pasolini's *Salò*.

Yet another connection between the works can be seen in *The Tempest*'s and *Salò*'s shared interest in the status of ceremony, culture, and ritual. *Salò*, of course, travesties rituals such as the marriage ceremony, while *The Tempest* features a highly ritualistic masque sequence intended to induct two young lovers into a sexual and marital union. It seems

notable, in this regard, that Jarman focused specifically on the ritualistic aspects of the play when preparing his adaptation, even commenting on its likely staging as part of the wedding celebrations for Elizabeth Stuart and the Elector Palatine, Frederick V. As much of Jarman's œuvre shows, he was deeply unsettled by the loss of the traditional, ritualistic elements of European culture. He had a "habit of looking back to a lost Elizabethan arcadia," and he railed against the Conservative policies of Margaret Thatcher for not "actually conserving anything" (Wymer 2005, p. 6). This streak tallies, in many ways, with Pasolini's own hatred of the flattening power of late capitalism, which he felt had replaced Europe's cultural heritage with a bland and all-encompassing consumerism, and we see this in the way *Salò* remolds cultural history (in the form of modernist paintings, philosophical quotations, and more) purely for the enjoyment of those in power. As Olga Solovieva (2019, p. 69) argues, the film represents "a concentration camp as a museum of modern art."

The overlapping themes of ritual and isolationism in both *The Tempest* and *Salò* also seem to have informed another link that Jarman identified between the two works: namely, a connection between the secretive discourse of Renaissance magic and contemporary forms of queer coding, both of which, he noted, relied on closed, hermetic ceremonies and hidden spaces:

> Part of my interest in the magician John Dee was his preoccupation with secrets and cyphers... Why this obsession with the language of closed structures, the ritual of the closet and the sanctuary? the prison cells of Genet's *Un Chant d'amour*, the desert encampment of *Sebastiane*; Angel, insulating himself with magick, screening himself off; Cocteau's *Orphée*, an attempt to steal through the screen into the labyrinth and usurp the privileges only the cabal of the dead may confer; the wall of unreality that girds the house in *Salò* and its victims who are told: What is about to take place here will have never happened, you are already dead to the world outside. (Jarman 1987, p. 66)

Jim Ellis (2001, p. 277) argues that this interest in secrets and cyphers suggests an explicit connection between the early modern discourse of alchemy, and the modern discourse of camp, claiming that "camp is perhaps the modern equivalent of alchemy, practiced by a marginalized group, dependent upon a specialized knowledge, and representative of an entire philosophical outlook." In this analysis, the world of Jarman's *Tempest* becomes a safe space for pursuing the otherwise marginalized and mistrusted practices of both alchemy and camp. Ellis argues that "John Dee's magic is for Jarman a metaphor for an oppositional gay filmmaking tradition" (Ellis 2009, p. 64). However, this clearly depends on a particular interpretation of the role that magic, secrecy, and hermeticism play in Jarman's imagination. For while it is true that Jarman does construct a genealogy linking gay filmmaking with Renaissance magic, it is less certain that this is valued for its "oppositional" or even its redemptive promise. Indeed, it seems notable that Jarman, in referencing *Salò* in his writing here, focuses specifically on the moment, mentioned above, when the narrative transitions from the exterior to the interior, when the victims enter the progressively more restrictive circles of absolute power and control. That theme, moreover, runs through the other films he cites, with Cocteau's *Orphée* (*Orpheus*, 1950) recalling yet another descent into hell and Jean Genet's *Un Chant d'amour* (*A Song of Love*, 1950) depicting homosexuality in the confines of a prison. Arguably, this theme suggests another interpretation of the magic in Jarman's *Tempest*, and Michael O'Pray (1996, p. 112), echoing Colin MacCabe (1996, p. 194), has argued that the film's magical elements should instead be read as a "comment on the Elizabethan 'police state.'"

The idea of *The Tempest* as a prison, or as an inescapable labyrinth in which viewers find themselves trapped, obtains in Jarman's thinking about the play from the very first drafts he made for his adaptation. His first version of the screenplay from 1974 begins: "SCENE: PROSPERO's palace, a prison room," clearly eliding Prospero's kingdom with a cell, and the pages of this draft are covered with labyrinthine designs and inescapable mazes.[2] More specifically, Jarman originally imagined the film as one in which the "whole is

enacted in Prospero's mind," a conceit that was to be established by Prospero "dubbing...his visitor's voices" ("Screenplay": fol. 2). This technique would, of course, later be realized separately in Peter Greenaway's 1991 adaptation, *Prospero's Books*, but the idea had clearly occurred to Jarman many years before. Moreover, his mechanism for accomplishing this depended on using the vocal track of the film to create magical, cinematic effects. From the outset, therefore, we might say that Jarman envisaged the film's soundscape as a means of including the audience within the prison of Prospero's mind.

This idea is still present in Jarman's notes for the final version of the film, in which he describes Stoneleigh Abbey as "an enclosed world, almost a nightmare."[3] There, however, that sense of enclosure is more concretely connected to Jarman's understanding of Renaissance magic, which is also linked to twentieth-century psychodynamic theory:

> In *The Tempest* I hope to recapture something of the mystery and atmosphere of the original without descending to theatrics. There are films where magic works. Like *The Cabinet of* [*Dr.*] *Caligari*, a mesmeric, alienating world of asylums, deserted rooms, and sleepwalkers which is half explained, or Bergman's *Hour of the Wolf* in which the whole landscape is alive with apparitions and malevolent dreams. It is therefore vitally important to see the play in the light of Freud and more particularly Jung as without the use of equivalents it can never hope to work for a modern audience with anything but a distant echo of its real power. ("Script": fol. 14)

This understanding of Renaissance magic through the lens of psychoanalytic theory makes sense, given that, in the twentieth century, it was Freud and Jung who most forcefully asserted the relevance of the study of dreams (previously considered either magical or trivial phenomena) as a mechanism for understanding the unconscious desires and drives of individuals. Jung, moreover, took the study of mystic topics such as dreams and alchemy even further than Freud, and there are quotes from Jung and the mystics he cited adorning Jarman's notebooks for *The Tempest*.[4] Importantly, however, what Jarman's notes suggest is that magic and alchemy in his film were not intended to function merely analogically (for example, by standing for a particular marginalized community or for the discourse of camp); rather, Jarman connected them physically or affectively with psychological experiences, with the emotional transferences that take place between people. Key to his approach, in other words, was an effort to construct a film in which, as in the other films he mentioned, magic would *work*, in which it would continue to hold a "distant echo of its real power."

### 3. An Enduring Question Mark

If Jarman did want the magic in his film to function as part of a dark hellscape redolent of Pasolini, Bergman, and expressionist cinema, then it is also the case that he saw the masque sequence as a break from this. Indeed, in his early drafts for the film, he imagined that while the majority of the picture would be in "black and white, shot like a German expressionist horror film (*Nosferatu*)," at the end it would "burst into radiant color...transformed to a glittering diamond vision."[5] This was in keeping with how he viewed the masque in Shakespeare's play:

> *The Tempest* is a masque; what it lacks, in the theatre productions I've seen, is a sense of fun. You wouldn't subject a sixteen-year-old at her wedding to an indigestible evening. Whatever is buried in the play, the surface of it must glitter and entertain. (Jarman 1991a, p. 183)

Entertain it does. Jarman's version of the masque, in which Elisabeth Welch "replace[s] Iris, Ceres and Juno" with her sumptuous rendition of the blues tune "Stormy Weather" is undoubtedly spectacular (Jarman 1991a, p. 183). It is perhaps for this reason, due to the sharp contrast between the masque and the rest of the film, that critics have been tempted to view Welch's performance as the "final triumph" of Jarman's film (Ellis 2001, p. 279). To a certain extent this is true, but as Steven Dillon points out, even in this joyous celebration, it

is difficult not to hear the deep sentiments of trouble and regret that flow through Elizabeth Welch's lyrics. These lyrics "conflict with the marriage celebration and even with the song's performance. Welch is splendid and smiling, radiating and golden in her costume, but the blues song is concerned with what happens when 'my man is away'" (Dillon 2004, p. 97; see also Harris and Jackson 1997). Jarman himself pointed to this ambivalence in the ending of *The Tempest*, noting that: "after a finale of merriment and reconciliation, the humans in the tale board the ship, sailing away into a sunrise where everyone will live happily ever after. Or will they?" ("Promotional Material": fol. 2). In his film, Jarman leaves the question of resolution open, suggesting that even if the surface of *The Tempest* does "glitter and entertain," that does not completely redeem the darkness that is buried within. Jarman's commentary on the play, written as part of the "Promotional Material" for his finished version of the film, elaborates on this at length:

> *The Tempest* is the last of the great plays of William Shakespeare. Traditionally— for convenience—placed amongst his Comedies, it is in fact nothing of the kind. *The Tempest* is a play of contradictions, a beguiling mixture of "sweet music" and jarring discords, of ugliness and beauty, magic and the mundane. Within *The Tempest* romance and retribution exist in equal measure, fantasy merges into nightmare, reality and illusion collide. A free spirit is slave-bound with gossamer threads crueler than chains; out of the foul mouth of a monster comes pure poetry. *The Tempest* belongs to no category, save its own. Its quality is elusive: an enduring question-mark which has passed down the centuries. ("Promotional Material": fol. 1)

Jarman views the play, therefore, much as he views magic in the early modern period, as a "question-mark," and he allows its ambivalences to run right the way through his film. Indeed, aside from the relatively short masque sequence, the majority of the film is dominated by nightmarish, rather than dream-like, representations of magic, emphasizing what Colin MacCabe calls "Prospero's reign of terror" (MacCabe 2011, p. 506). This is signaled in several earlier scenes in which Prospero generates a series of twisted parodies of the later masque. In one of those sequences, as mentioned, Prospero and Ariel are shown terrorizing Caliban, Stephano, and Trinculo (who are engaged in a camp and drunken revelry of their own), by attacking them with the howls of phantom dogs. Following this, Prospero orchestrates a further performance piece, in which people of short stature clothed in oversized Elizabethan dresses carry out a strange and aggressive dance to torture the prisoners. William Pencak astutely remarks that these individuals resemble the figures in Velázquez's *Las Meninas* (an intertext also highlighted by Harris and Jackson 1997, p. 90) and suggests that the scene highlights the dark "underside of court life" (Pencak 2002, p. 105). In the film, it is clearly a nightmarish foreshadowing of the sailors' dance in the masque and, in our context, even recalls some of the grotesque rituals and ceremonies represented in Pasolini's *Salò*. Incidentally, on this point, Gian Maria Annovi (2012, p. 166) has argued that elements of *Salò*'s aesthetics also seem to have been "inspired by Velazquez's masterpiece," notably the narrative performance pieces in the "Circle of Manias," which echo the composition of the seventeenth-century painting. The question of whether Pasolini or Jarman intended these overlaps is uncertain, but they certainly form a suggestive visual network, inflecting Jarman's film with the intertexts of both Pasolini and Velázquez.

These Pasolinian echoes mean that the abuses of Stephano, Trinculo, and Caliban, along with the rest of Prospero's prisoners, cannot help but linger beneath the surface of the masque in Jarman's film, suggesting that magical power (and indeed power in general) works in multiple directions. It has the capacity to isolate as much as to bring together, a fact that is evidenced by the treatment of Caliban, Stephano, and Trinculo when they themselves arrive at the masque. Though their costumes are no more elaborate or ostentatious than those of Miranda and Ferdinand at this point, when these companions burst on to the scene, they elicit nothing but laughter and scorn from the young couple and their guests. Subsequently, Prospero chastises Caliban for wanting to be "king of the isle" (5.1.288) and tells him to "go to" (5.1.293). Though the trio remain present for the rest of the

performance, they are undeniably marginalized, cast off to the side and relegated to the status of onlookers. If a "brave new world" is predicted by this elaborate masque sequence, then Jarman seems to imply that it is a world for some and not for all.

These more troubling undertones of the masque also remind us that, though the masque may well be the film's most memorable sequence, it is not where it ends. Instead, the film ends with the masque's aftermath, almost another travestying. In this final scene, we see Prospero slumped in his chair asleep (as he was for the storm at the beginning of the film), still occupying the same room in which the masque took place. Now, however, he is surrounded by the detritus from that earlier celebration. The gold and glitter of the spectacle have faded to a gloomy grey, and the characters have all departed, save for Prospero and Ariel. In this quiet, somber atmosphere, Ariel waits expectantly, clearly hoping that the magician will offer him the freedom that he "promised/Which is not yet performed" (1.2.243–44). That freedom, however, is never granted, and Ariel ultimately tip-toes nervously past the sleeping Prospero and vanishes in the film's final moment of magical trickery, though, it is arguably only now, after this last moment of cinematic illusion, that the audience is made to understand the function of magic in Jarman's film. Following Ariel's departure, the camera moves closer in, filling the screen with Prospero's face. As he sleeps, his lips not moving, we hear Heathcote Williams speak in voiceover, closing out the film with Prospero's immortal words which, in the play, form the end of the masque:

> We are such stuff
> As dreams are made on, and our little life
> Is rounded with a sleep (4.1.156–8)

With this ending, Jarman's *Tempest* becomes visually, sonically, and formally "rounded with a sleep." Sleep is where we began, with Prospero in bed during the opening shots of the storm, and it is where we end up. Importantly, though, this is not just Prospero's sleep; as an audience we are incorporated within it. The boundaries between interior and exterior fold into one another here. Steven Dillon has drawn attention to this process, arguing that in this final scene "the clear division between interior house (the cerebral realm of Prospero) and exterior beach and ocean is now collapsed, since the house is strewn with leaves. The inside is an outside" (Dillon 2004, p. 97). In other words, Dillon suggests that we are brought into the folds of Prospero's dream. This process is also reinforced sonically by the way in which the sequence collapses the distinction between diegetic and extra-diegetic sound. The fact that Heathcote Williams's voiceover is heard while an image of him not speaking appears on screen encourages the idea that we have somehow entered Prospero's mind and are sharing the space of his dream. In this ambiguous film space, we are not quite sure whether we are inside Prospero's head, or inside our own. We become trapped, in a sense, in Prospero's prison as, significantly in Jarman's version, Prospero remains alone in his isolated world within the house and does not sail away with the courtiers to "live happily ever after." Such a prison seems characteristic of the overlapping worlds of magic and the unconscious mind that Jarman drew attention to in his notes. It is also, though, highly reminiscent of the ending of *Salò*.

As mentioned above, Jarman viewed Pasolini's film as one in which the "private" sphere of "pure relations" collapses due to a "line of betrayal." Such a betrayal also characterizes the film's final scene, at the end of the "Circle of Blood." During this section, the film's four libertines take turns occupying a chair overlooking a window, and we watch them watching a series of horrific acts of torture that are enacted upon their victims. As an audience watching *Salò* here, we are being invited to separate ourselves from the action of its final sequence, for Pasolini makes the act of spectating decidedly alienating. Whereas the majority of the film revolves around the theatrical narration of graphic and erotic stories in a shared auditorium for (ostensibly) the pleasure of an audience, in this final scene, we observe the action through multiple intermediary layers. Our entire frame of vision, for example, is constrained by the binoculars through which the libertines watch the acts of violence, and we are separated from those acts by the further remove of the grated bars on

the window. We are even dislocated from the action by way of the soundscape. Although we can see the acts of violence on screen, we hear no cries of pain or sadistic laughs. All we hear are the eerie tones of Carl Orff's *Carmina Burana*, which haunts the background of the scene. As with Jarman's *Tempest*, however, the film does not end with this memorable spectacle. In the film's ending, Pasolini cuts away from the shots of violent torture taking place outside to a young boy who is sat on the floor with his rifle. He is not watching the events, but he reaches up and toggles the dial on a radio beside him, at which point *Carmina Burana* cuts out, and the title tune of the film, composed by Ennio Morricone, which had initially played over the opening credits, cuts in. The boys in the room get up and dance the film away into a dreamy fade to white. The result is twofold. Firstly, this link with the title music has the effect of cycling the film back into its beginning, suggestive of yet another all-encompassing and Dantesque circle. Secondly, though, and similarly to Jarman's film, it also collapses the divide between diegetic and extra-diegetic sound. When we first hear Carl Orff's haunting melody, we assume it is extra-diegetic; the film gives us no reason to think otherwise. We have seen no choral singers, observed no flautists in the palazzo, and the only musician present in the villa has already committed suicide by casting herself from an open window. We are led to assume, therefore, that the sounds we are hearing belong in some way to a world beyond the screen, beyond the spectacle of on-screen violence. However, at the moment when the young boy switches over the radio, we realize that what appeared as extra-diegetic was nothing of the sort. I would argue that, in this moment, Pasolini leaves us with nowhere to go. What we had taken for the internal world of the film is collapsed together with what we had taken for the external world that we inhabit as an audience. We are forced suddenly to recognize not the separation but the contiguity between ourselves as audience members and *Salò* as spectacle.

As is well known, Pasolini in *Salò* sought to make a film that would be, as he put it, "as inconsumable as possible" (Bondavalli 2010, p. 408). He aimed, as Patrick Rumble argues, to accomplish the "death of cinema," to ensure that cinema passed beyond the bounds of watchability (Rumble 2004, p. 159). And this occurs here in a very particular way: Pasolini ensures that cinema can no longer simply be viewed and left behind. Even as the young boys in the closing sequence seem to invite us to dance away the memory of what we have just witnessed, we are forcibly reminded that the established contract between actors and audience has been ruptured. The elements that previously divided the on-screen world from the off-screen world have been dissolved, and even though we might decide just to turn off our screen, or to walk out of the movie theatre, Pasolini reminds us that this would merely be tantamount to switching over the channel.

For this reason, it could be argued that the final moments of Pasolini's and Jarman's films are in some ways a radicalization and reformulation of the process of transitioning between the interior and the exterior of the cinema-space, as outlined by Roland Barthes in his famous essay "Leaving the Movie Theatre" (Barthes 1989). It is not clear whether Jarman (or Pasolini) ever read this essay, but as mentioned, Barthes was one of the authors whom Jarman recalled when he himself exited the cinema after watching *Salò* in February 1990, so it seems a fitting comparison. In this essay, Barthes describes a cinema-goer (possibly himself) who leaves the theatre after having been "glued to the representation," and who

> walks in silence… a little dazed, wrapped up in himself, feeling the cold—he's sleepy, that's what he's thinking, his body has become *soportif*… In other words, obviously, he's coming out of hypnosis. And hypnosis (an old psychoanalytic device—one that psychoanalysis nowadays seems to treat quite condescendingly) means only one thing to him: healing. (Barthes 1989, p. 345)

It is, I would argue, precisely this process of psychoanalytic "healing" that the endings of *Salò* and Jarman's *Tempest* deny. Indeed, perhaps it was precisely this denial that provoked Barthes himself to argue famously that, while *Salò* was a "flop of figuration" and failed absolutely as an "analogy" (because "fascism is too dangerous and too insidious a danger to be treated by simple analogy"), it still managed to function somehow "on the level of affect" (quoted in Allen 1982, p. 102). It was, in fact, its affective power that made it

impossible for anyone, as Barthes put it, to "redeem it." I would argue, in similar terms, that Jarman's *Tempest* also functions primarily "on the level of affect" by incorporating its audience within the circle of its magical dreamscape. In this way, by making us aware of the continuity, rather than the disjunction, between the magic of cinema and the waking world in which we live (by collapsing one into the other), both Pasolini in *Salò* and Jarman in *The Tempest* ensure that Barthes's cinema-goer no longer has the opportunity to transition out of hypnosis and into the world beyond. They remain, in a sense, captivated; there is no release from the magical spell of the cinema and the closed structures that it represents. This, then, is perhaps how Jarman ultimately tried to link together theories of Renaissance magic and the sadism of Pasolini's cinema and weave them both into a reimagining of *The Tempest*. Like Pasolini, he endeavoured to collapse the boundary between interior and exterior, incorporating us within Prospero's cinematic dream and making it impossible for us simply to dissociate ourselves from the final Dantesque circle: the circle of cinema.

Even more interesting, however, is the fact that this approach is, in many ways, not very far removed from that taken by Shakespeare himself. In the final epilogue to Shakespeare's play, after the so-called resolution that Prospero masterminds within his own magic circle—the reclaiming of his dukedom, the marriage of his daughter, the unmasking of Stephano, Trinculo and Caliban—there follows a profound re-orientation of the space of theatrical performance. After Prospero releases his prisoners, he addresses the audience directly, asking them to release him from his "bands/With the help of your good hands" (5.epi.9–10): "As you from crimes would pardoned be," he says, "Let your indulgence set me free" (5.epi.9–20). Thus, what seems to be shaped in this final moment is a new kind of performative space, one from which, as with *Salò*, the audience finds it difficult to escape. Prospero's performance (which, it should be noted, has involved acts of torture, imprisonment, forced labor, sexual repression, and more) is shown to hang here on a power that rests in the audience's hands, in their power to applaud. In a way that seems remarkably similar to the ending of *Salò*, the moment shatters the tacit contract that exists between audiences and actors (or perhaps more accurately between victims, torturers, and spectators). Prospero, just like Pasolini, gives us no way out, even as he seems to offer us one, and the audience is immediately caught in the finely crafted web of his double bind. Either they refuse to clap, in which case they prolong, potentially indefinitely, Prospero's imprisonment, and thereby enact revenge upon him for the acts they have seen him undertake, or they applaud and endorse what they have seen, redeeming Prospero and setting him free. Thus, the ambiguities and ambivalences of *The Tempest* (what Jarman referred to as the "enduring question mark" of the play) are felt most palpably in this final moment. The audience is left with only the options that Prospero affords them: they can applaud, and thereby forgive him, or they can suspend their applause and keep him confined for his crimes.

Prospero's epilogue, then, just like the conclusion of Pasolini's film, suggests that an audience is trapped, held captive within the all-encompassing experience of spectatorship. Across the centuries, these texts seem to say to us: feel free not to applaud; feel free to scream your disgust and outrage at what you have seen; feel free to stand up and walk out of the cinema or the theatre; but none of these responses will absent you from the circle, be it a circle of mania, shit, blood, magic, or even cinema itself. Such a response would merely be tantamount to switching off the radio, as the young boys do at the end of *Salò*. In this way, even though Jarman's film does not end with this Shakespearean epilogue, its concluding scene reproduces the affective or phenomenological experience on which it hangs. It seems, therefore, to link an understanding of the affective power of Renaissance magic with the psychosocial experience of cinema, providing a peculiarly Pasolinian twist on a mystical Shakespearean play, one from which there is ultimately no escape.

**Funding:** This research received no external funding.

**Institutional Review Board Statement:** Not applicable.

**Informed Consent Statement:** Not applicable.

**Data Availability Statement:** Not applicable.

**Acknowledgments:** I would like to acknowledge the kind support provided by the BFI Special Collections Archive, which provided access and guidance to the Derek Jarman Collection.

**Conflicts of Interest:** The author declares no conflict of interest.

## Notes

[1]  Archival materials are quoted from the Derek Jarman Collection in the BFI Special Collections Archive, London. Derek Jarman, "Script. A TEMPEST. A Film Treatment of Shakespeare's Play by Derek Jarman and James Whaley," (1976–1979?), Box 5, Item 4, Fol. 14, Appendix I, The *Tempest* (hereafter: "Script").

[2]  Derek Jarman, "Screenplay of *The Tempest*," (1974), Box 4, Item 5640 1A, Derek Jarman Collection, BFI Special Collections, Fol. 1 (hereafter: "Screenplay").

[3]  Derek Jarman, "Promotional Material" (1979), Jarman II, Box 28. Item 3: "Promotional Material" (hereafter: "Promotional Material").

[4]  Among many examples, on Fol. 16r of an elaborate notebook compiled for *The Tempest*, dated August 1975, we find alchemical drawings by Jarman accompanied by words from Michael Maier's *Scrutinium Chymicum* (1687) written vertically down the left hand side of the page: "Sol et eius umbra." This is also quoted in C. G. Jung, *Dreams* (London: Routledge), 113. See Derek Jarman, "The Tempest" Notebook (August 1974), Box 28, Item 1, Derek Jarman Collection Fonds, BFI Special Collections.

[5]  Derek Jarman, "Production Note for 'A Tempest,'" (Project Year, 1979), Item 5640 1C, Derek Jarman Collection, BFI Special Collections Archive.

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
