# Peer review of "Derek Jarman’s Tempest, William Shakespeare’s Salò"

_humanities, doi:10.3390/h12040076_

Round 1
Reviewer 1 Report
This was a good essay which addressed directly Derek Jarman’s apparently puzzling reference to The Tempest as ‘Shakespeare’s Saló’ by detailing Jarman’s lifelong fascination with Pasolini’s films, including his particular interest in Saló, and then establishing a number of connections between Jarman’s film of The Tempest and Pasolini’s final film which was based on de Sade’s 120 Days of Sodom. The essay used some appropriate archival material (much of which has become fairly familiar to Jarman scholars) and was very clearly written. The concluding section of the essay on the endings of the two films was particularly strong and the essay could be published without much revision. However, it could also be strengthened a bit.
The essay correctly says that much of the criticism of Jarman’s Tempest has concerned itself with whether or not the film is politically ‘progressive’, something which the link with Saló renders immediately problematic. But there is no reference to critics who have taken a more aesthetic and psychological approach, such as Diana Harris and Macdonald Jackson’s 1997 essay or Rowland Wymer’s 2005 book. If the author had consulted the latter, he/she would have picked up a number of details which would have strengthened the argument. For instance, I didn’t notice any reference to the flashback scene in which Sycorax is shown controlling Ariel with a collar and chain, surely relevant to the Saló connection which is being made. In some of Jarman’s draft notes there is a more extreme and even more obviously sadomasochistic version of this scene described. Another relevant piece of information in Wymer’s book is that Jarman also compared his later film The Last of England to Saló and in it referenced the last scene of Pasolini’s film with a shot of two soldiers dancing together.
More generally, I would have liked to see more acknowledgement that Jarman had a fascination with sexualised violence that is apparent in many of his films, particularly Sebastiane, Jubilee, The Garden, and Edward II. It seems odd to talk about imprisonment and sadomasochism in The Tempest (where much of it is below the surface) and not mention the very explicit treatment of these topics in Jarman’s other adaptation of a Renaissance play, Marlowe’s Edward II. The ‘darker and less forgiving model of sexuality’ which Jarman responded to in Pasolini was one strand in his own complicated psychology. One cannot discuss Prospero’s exercise of power in Jarman’s film without also mentioning Jarman’s very strong identification with him (which is why he is shown as a much younger man than is normal in stage productions). Jarman also was fascinated by the way Pasolini, when directing Saló, put his cast through every conceivable form of simulated degradation whilst walking among them clad in ‘immaculate suits’. One of the reasons Jarman and Pasolini are great film makers is that they were not afraid to confront uncomfortable aspects of their own and other people’s sexualities (‘The great queer artists deal in negatives’, as Jarman wrote).
Some more minor points:
Describing the possible link between Shakespeare’s play and Pasolini’s film as ‘last works’ as ‘pseudo-biographical’ (146) seems too strong. There is a fairly strong scholarly consensus that The Tempest was Shakespeare’s last solo-authored play, being succeeded only by the collaborations with Fletcher. Much about The Tempest seems to be a self-conscious gathering together of themes which have resonated throughout Shakespeare’s career and a prolonged self-reflection on the role of the artist and the power of art. In other words, it was written as a ‘last play’, just as Bergman’s Fanny and Alexander was planned as a ‘last film’, even though, technically speaking, it was followed by After the Rehearsal and Saraband.
I would be wary of accepting as a fact Jarman’s belief in an ‘epistemological shift that occurred in England in the early 1600s’ (82-3) which left Shakespeare as a belated Elizabethan looking back nostalgically to the culture of his youth. I think there is plenty of evidence that Shakespeare was actually more positive about James I than he was about Elizabeth (almost alone of major poets, he did not write an elegy for her in 1603, something which was noticed at the time).
There were very few typographical errors. The three I noticed were:
93 Ellie for Ellis
222 Tempst for Tempest
516 flortist for flautist
Author Response
Thank you very much for the exceedingly helpful comments, as well as for the recommendations for further reading that I had overlooked. They proved very helpful for revising the article. I have tried to respond to all of the points made, and the relevant areas have been highlighted in the document:
- Harris and Jackson's essay has been consulted and helpfully included to bolster the analysis of Jarman’s psychological approach to The Tempest (see 277, 298, 419).
- Similarly, Wymer's chapters on The Tempest have also been consulted to provide some very helpful background and bolster the argument. Reference to the flashback scene (which was actually present in an earlier draft but was cut due to the word count) has been made and helpfully elaborated with reference to Wymer (see 301-304).
- Several other very useful details pointed out by the reviewer, such as the reference to the influence of Salò on The Last of England (which I certainly overlooked) have been added (see 143).
- An acknowledgment of Edward II's darker sexuality has been made, but I have found it impossible to venture into a more detailed discussion of Edward II given the already rather packed article. I hope this will be sufficient to pay reference to this important and interesting adaptation.
- The helpful comment regarding Jarman, Pasolini, and Geney being artists who "deal in negatives" provided a particularly useful development of the argument, and even afforded a restructuring of the first section of the essay to try and elaborate more clearly the influence of Pasolini's "darker and less forgiving model of sexuality" on Jarman's work. The hope is that this now provides a slightly more coherent assessment of this line of influence.
- The term "pseudo-biographical" has been removed (see 221), and this section has been slightly rewritten to account for the well-made points about what does or does not constitute a "last" work. Hopefully the current paragraph now provides a slightly more nuanced account.
- An interjection has been included to note that it was Jarman rather than necessarily Shakespeare who saw the Jacobean period as a definitive break from the Elizabethan (the small note: "in his mind" has been added at 83).
- The typographical errors have been remedied.
Thank you again for such a generous and detailed review.
Reviewer 2 Report
See attached

Author Response
Many thanks for such a generous review, with helpful comments for bolstering the argument. I consulted the article by Annovi, which was very interesting in itself, and I included a note on the overlap between Pasolini and Velázquez at the relevant point in the article (454). I also revised the paragraph on "the status of ritual" to provide more detail on the significance of this for both Jarman and Pasolini, pointing to their shared interest in cultural heritage and tradition. Hopefully this has helped to clarify the contemporary political resonances of ritual for Jarman.
My sincere thanks again for the suggestions and the review.